# A Review on Barrier Properties of Poly(Lactic Acid)/Clay Nanocomposites

**DOI:** 10.3390/polym12051095

**Published:** 2020-05-11

**Authors:** Shuvra Singha, Mikael S. Hedenqvist

**Affiliations:** KTH Royal Institute of Technology, School of Engineering Sciences in Chemistry, Biotechnology and Health, Department of Fibre and Polymer Technology, SE-100 44 Stockholm, Sweden

**Keywords:** barrier properties, poly(lactic acid), clay, nanocomposite, permeability

## Abstract

Poly(lactic acid) (PLA) is considered to be among the best biopolymer substitutes for the existing petroleum-based polymers in the field of food packaging owing to its renewability, biodegradability, non-toxicity and mechanical properties. However, PLA displays only moderate barrier properties to gases, vapors and organic compounds, which can limit its application as a packaging material. Hence, it becomes essential to understand the mass transport properties of PLA and address the transport challenges. Significant improvements in the barrier properties can be achieved by incorporating two-dimensional clay nanofillers, the planes of which create tortuosity to the diffusing molecules, thereby increasing the effective length of the diffusion path. This article reviews the literature on barrier properties of PLA/clay nanocomposites. The important PLA/clay nanocomposite preparation techniques, such as solution intercalation, melt processing and in situ polymerization, are outlined followed by an extensive account of barrier performance of nanocomposites drawn from the literature. Fundamentals of mass transport phenomena and the factors affecting mass transport are also presented. Furthermore, mathematical models that have been proposed/used to predict the permeability in polymer/clay nanocomposites are reviewed and the extent to which the models are validated in PLA/clay composites is discussed.

## 1. Introduction

Amid the growing environmental concern about the decreasing fossil resources and the increasing plastic footprint, biopolymers obtained from renewable resources such as agricultural products represent a promising alternative to the non-degradable petroleum-based polymers for short-life range applications, for example food packaging [1,2,3,4,5]. Poly (lactic acid) (PLA) has emerged as the frontrunner among the many biopolymers in this regard owing to its many eco-friendly attributes such as low energy consumption during production, availability and low cost of the raw material, biodegradability in soil and water and being non-toxic to the environment [6,7,8,9,10,11,12]. Although the most successful application of PLA is in the containers and food packaging industry, other applications include biodegradable scaffolds for tissues, bioresorbable implants, surgical equipment, intravenous administration of antivirals, cardiovascular stents and controlled drug delivery. PLA is also used for making fibers in the textile industry and mulching materials for agriculture. The important properties which make PLA a promising candidate for food packaging is that it possesses sufficient thermal stability, i.e., the onset degradation temperature lies in the range of 330–350 °C [13,14] and good mechanical properties: tensile strength of ca. 50–70 MPa, Young’s modulus of ca. 3 GPa, elongation at break of ca. 4% and impact strength of around 2.5 kJ/m^2^, making it a useable substitute for single-use plastics such as PE, PP and PET [10,15].

PLA is derived from renewable agro-resources such as corn, cassava, potato, cane molasses and sugar beet, and hence is considered as an eco-friendly thermoplastic. The polymer is produced from the monomer of lactic acid (LA), the simplest hydroxy acid, which is obtained either biologically by the fermentation of carbohydrates by lactic bacteria belonging to the Lactobacillus genus or by chemical synthesis [16,17,18]. PLA is produced through two important routes—(a) direct polycondensation (DP) of LA and (b) ring opening polymerization (ROP) of the cyclic dimer of LA, i.e., lactide (Figure 1B). The DP route is an equilibrium reaction which demands high temperature, long reaction times and continuous removal of water from the reaction vessel, often leading to low molecular weight PLA [19]. Hence, ring opening polymerization (ROP) of lactide (cyclic dimer of LA) in the presence of a robust catalyst–initiator, tin(II)bis(2-ethylhexanoate) (Sn(Oct)_2_) and alcohol, is the preferred synthesis route in industry, which can result in high molecular weight PLA [7,20,21].

LA is a chiral molecule and exists in two stereo-isomeric forms (optical isomers) l-lactic acid (l-LA) and d-lactic acid (d-LA). Two optically inactive forms are also available which are the meso-LA and racemic mixture (50:50) of l-LA and d-LA (Figure 1A) [16]. PLAs formed from the isotactic sequence of l-LA and d-LA are referred to as PLLA and PDLA, respectively. PLA prepared from a racemic mixture of both the enantiomers and from meso-LA is referred to as PDLLA [22]. The final properties of PLA largely depend on the ratio and distribution of the LA enantiomers in the polymer chains. A high L-isomer in the chains results in a crystalline matrix, whereas a high d-isomer (>15%) results in an amorphous matrix. The meso-form (atactic PDLLA) is also amorphous. The polymer chain orientation and packing affect the crystallinity, crystal thickness, spherulite size and morphology [12]. These are important factors which influence two important physical properties, i.e., mechanical and barrier performance [23,24,25]. Although PLA is among the best biodegradable and nontoxic polymers with high thermal stability and good mechanical stability (although with low extensibility without a plasticizer), its limiting property is its permeability to low molecular weight gases, vapors and organic molecules [26]. Permeation of oxygen and water vapor through polymer films can drastically decrease the service performance of a packaging material, thereby making it difficult to maintain food quality throughout its shelf life [27,28]. Research on mass transfer in polymers is, therefore, of high importance. Inclusion of two-dimensional (2D) platelets or disk-shaped nanoparticles in the polymer matrix has proven to be a good strategy to significantly decrease gas/liquid permeation in polymers. The 2D inclusions act as physical obstacles in the diffusion path of the permeant molecule creating a tortuosity effect. This helps to enhance the barrier performance of the polymer and increases the food shelf-life [29,30,31,32]. Nanoclays such as mica, saponite, montmorillonite and kaolinite are widely used 2D nanoparticles for improving the barrier properties in many polymers [33]. Thousands of publications can be found on PLA/Clay nanocomposites which have largely focused on improving the thermal [34,35,36], mechanical [37,38,39] and optical properties [40,41] and biodegradability [42,43]. However, only a meagre number of publications have been devoted to the study of mass transfer in PLA, and just a handful on PLA/clay nanocomposites. This encourages further review so as to update the trends, accomplishments and recurring challenges in this field. This review intends to highlight the usefulness of clay platelets for improving the barrier properties of PLA. First, the current methods of fabrication of PLA/clay nanocomposites are summarized, and then the barrier performance of the nanocomposites is reviewed. A brief introduction to the theory, mechanism and factors affecting mass transport in polymers is presented followed by a description of some of the important mathematical models that have been proposed to predict permeability in polymer/clay nanocomposites. The validation of some of the models in PLA/clay nanocomposites is reviewed.

## 2. PLA/Layered Silicate Nanocomposites

The 2D layered inorganic nanofillers like clays and silicates, owing to their abundance, low cost, high aspect ratio, rich intercalation chemistry, high strength and stiffness and thermal stability, provide favorable synergetic effects that help to significantly improve many polymer properties, especially mechanical and barrier properties [44,45,46,47]. However, the extent of dispersion of the clay layers and the morphology thus achieved in the polymer matrix (intercalation, exfoliation, mixed intercalation and exfoliation, aggregation, etc.) greatly affect the gas barrier properties. To achieve a high level of exfoliation and desired orientation of the platelets has remained a challenging task [48,49,50,51]. Good dispersion can be realized by increasing the affinity between the clay layers and the polymer through organic modification of the interlayer galleries with organic ammonium, sulfonium or phosphonium cations. A detailed and extensive list of common organic modifiers has been reported by Nordqvist and Hedenqvist [33]. The common routes to achieve dispersion of the organo-modified layered silicates in a PLA matrix are solution intercalation, melt processing and in situ polymerization (Figure 2) [52].

### 2.1. Solution Intercalation

Solution intercalation is one of the easiest techniques on a laboratory scale to prepare nanocomposites. In this technique, clay platelets are first exfoliated in a solvent in which the polymer is also soluble. The polymer solution is then mixed with the clay suspension, where the polymer chains intercalate/are adsorbed on the surface of the platelets and form a clay–polymer complex. The solvent is later removed by evaporation. This method is considered environmentally unfriendly because of the use of organic solvents [51]. Maharana et al. [12] demonstrated the preparation of PLA/clay nanocomposites using a solution intercalation method and showed improved mechanical and barrier properties of the nanocomposites. The effect of the structure of different organic modifiers of clay nanoparticles, Cloisite-15A, -25A and -30B modified with dimethyl dihydrogenated tallow quaternary ammonium, dimethyl hydrogenated tallow-2-ethylhexyl ammonium and methyl tallow-bis-2-hydroxyethyl quaternary ammonium, respectively, was studied by Pochan and Krikorian [53] to determine the extent of exfoliation of the nanoclay in a PLA matrix by solvent intercalation. Cloisite is a montmorillonite (MMT) clay: the term ‘Cloisite’ followed by an alphanumeric sequence refers to the commercially available clay, whereas the organically modified MMT (OMMT) refers to the tailor-made clay prepared by individual research groups. Cloisite 30B containing an organic diol in the inter-galleries established favorable interactions with the carbonyl functionality of PLA, leading to significant intercalation of PLA chains into the clay spacing. Hence, PLA/Cloisite 30B formed the best nanocomposites in terms of maximum intercalation.

### 2.2. Melt Intercalation

This is a widely used technique to fabricate PLA/clay nanocomposites. The method involves mixing organo-modified nanoclay and the polymer and heating the mixture above the melting temperature of the polymer, either under shear or no shear. Due to the high temperatures and mechanical forces used, polymer chains are forced to diffuse into the clay galleries, giving rise to either intercalated or exfoliated nanostructures depending on the amount of polymer chains diffused into the silicate layers [32]. The main advantage of the technique is the specificity for the polymer intercalation into the clays as there is no solvent in the system that can give rise to competing clay–solvent or polymer–solvent interactions [54]. Most of the PLA/clay systems prepared by melt processing have resulted in intercalated structures. To achieve further exfoliation, Sabet and Katbab [55] investigated the role of oligo(ε-caprolactone) as compatibilizer. Although the effort did not result in complete exfoliation, it did result in flocculation of the clay layers due to hydroxylated edge-edge interactions and, therefore, better parallel stacking of the layers. However, fully exfoliated nanostructures were achieved by Chen et al. [56] who performed a second time functionalization of Cloisite 25A using an epoxy containing organic modifier—(glycidoxypropyl)trimethoxysilane. Melt processing of the nanoclay with PLA yielded fully exfoliated nanostructures when the epoxy content in the clay was high (about 0.36 mmol/g). PLA nanocomposites with epoxy containing Cloisite 25A showed better mechanical properties than those of the unmodified Cloisite composites. Melt blending of PLA in the presence of nanoclay with other polymers have been reported by several authors [57,58,59].

### 2.3. In Situ Polymerization

This technique is the most effective to obtain well exfoliated clay platelets in the polymer matrix. First, the clay is swollen in a suitable monomer melt or monomer solution. Then, polymerization is carried out induced by heat or radiation or by pre-intercalated initiators or catalyst. During the polymerization reaction, polymeric chains are formed inside the clay galleries which force delamination of the platelets in the matrix. Melt intercalation of LA monomer in the clay galleries followed by in situ ROP of PLA was found to be an efficient route to prepare high molecular weight PLA composites. Here, the silicate inter-galleries are considered as ‘‘nano-reactors’’ yielding high molecular weight PLA. Cloisite-Na^+^, Cloisite 20A, Cloisite 30B and organo-modified montmorillonite (abbr. as OMMT) (modified with hexadecyltrimethyl ammonium bromide and dioctadecyldimethyl ammonium bromide) nanoclays were used to first form the LA monomer–clay intercalated mixture. The mixture was subjected to ROP using Sn(Oct)_2_ as the catalyst for 2 h at 120–180 °C. High molecular weight PLA composite, ca. 126,000 g/mol, was obtained [60]. Katiyar and Nanavati [61] demonstrated a novel solid-state polymerization route to prepare high molecular weight PLA using a two-step in situ ROP process. The PLA prepolymer was first synthesized via ROP inside the clay layers followed by solid-state polymerization at 150–160 °C. Another innovative approach using in situ coordination insertion polymerization was reported by Paul et al. [62], referred to as the ‘‘grafting-from’’ approach. In this method, aluminum oxide reactive species was first formed in situ by reacting triethylaluminum with hydroxyl groups of the ammonium cation (organic modifier) of Cloisite 30B. ROP of the intercalated monomer was then carried out at the site of active species in the presence of initiator and catalyst.

## 3. Barrier Performance

The effects of the type of organo-modifier used, clay volume fraction, aspect ratio and dispersion on the barrier properties of PLA/organoclay nanocomposite films have been investigated. The nanocomposite films demonstrated improved barrier properties compared to the neat PLA film. Gorrasi et al. [63] prepared PLA nanocomposites with Cloisite 30B by the melt blending method where PLA, clay and polyethylene glycol (PEG) as plasticizer and stabilizer were mixed together in a counter rotating mixer. Films were then produced by compression molding the blend. These authors also prepared nanocomposite by in situ polymerization of PLA from the monomer–swollen clay. Fully exfoliated nanostructures were obtained from the in situ process that had significantly lower water solubility and diffusivity than the melt processed composite. It was also observed that the water vapor zero concentration diffusivity measured at 30 °C was decreased in the case of in situ blends by about two orders of magnitude compared to that of the melt-processed blend and neat PLA films.

The oxygen permeability of PLA nanocomposites prepared with different clay modifications (Cloisite 25A, OMMT modified with dodecyltrimethyl ammonium cation and OMMT modified with hexadecylamine) was investigated by Chang et al. [64]. The permeability of all the composites was found to be less than that of pure PLA films and, at a clay content of 10 wt%, the permeability decreased to less than half the value of the neat PLA film. The barrier performance was determined from the barrier improvement factor (BIF) which is the ratio of transmission through the neat film to the transmission through the composite film. The BIF values are tabulated in Table 1. Maiti et al. [65] studied the effect of chain length of organic modifier in different types of clay, smectite, mica and OMMT. The clays were modified with a phosphonium ion containing three butyl branches and an alkyl chain the length of which was varied from 1–16 carbon atoms. The composite containing the C16 modifier was only evaluated for oxygen permeability. Smectite clays showed better dispersion than other clays and, therefore, showed better barrier properties than the mica and MMT loaded films (BIF for the highest clay loading of 4 wt% is shown in Table 1). The mica system with stacked clay layers exhibited poor barrier performance and low modulus.

Ray et al. [66,67,68,69,70] measured the oxygen permeability of PLA/organo-modified clays in a series of papers where the nanocomposites were prepared by melt extrusion in a twin-screw extruder followed by compression molding the granulated material. In Ref [66], three different clays with three different organic modifiers were used to assess the PLA nanocomposite properties. MMT was modified with octadecyl ammonium and octadecyltrimethyl ammonium cations, saponite was modified with hexadecyltributyl phosphonium cation, and synthetic fluorine mica (SFM) was modified with dipolyoxyethylene alkyl(coco) methyl ammonium cation. Although the saponite system showed the best dispersion, higher barrier properties were obtained with the mica system. Synthetic fluorine mica modified with hydroxy functional ammonium cations was dispersed in the PLA matrix, resulting in intercalated stacks and a fairly large number of exfoliated layers being revealed in TEM micrographs. The highest barrier performance in terms of oxygen permeability was seen at 10 wt% mica content [67]. OMMT modified with octadecyltrimethyl ammonium cation [68] and also with the linear analog (octadecyl ammonium cation) [69], was blended with PLA. Interestingly, the linear surfactant-containing composite displayed higher barrier properties at 7 wt% clay than with the trimethyl functional surfactant.

Investigation of the effect of different processing parameters and techniques such as compounding and blown-film processing using a co-rotating twin screw extruder, were carried out by Thellen et al. [71]. The nanocomposite films showed 48% improvement in oxygen barrier and 50% improvement in water vapor barrier properties compared to the neat PLA film. Their results indicated that barrier property enhancements can be achieved by conventional processing techniques.

Lagaron et al. [72] used food contact approved nanoclays and amorphous polylactic acid (aPLA) to fabricate aPLA/organoclay nanocomposites. The organoclays used were Nanoter C1 (kaolinite) and AE21 (MMT) from nano-biomatters S.L. (Spain) (organo-modifiers not disclosed). TEM analyses revealed a mixed morphology of exfoliation and agglomeration of the clay platelets in the OM-kaolinite/aPLA matrix. The oxygen permeability BIF was higher (1.8) for the kaolinite system at 40% RH and 21 °C, whereas for the MMT system BIF was 1.1 (Table 1). The nanocomposites displayed very little swelling in water compared to the neat unfilled aPLA.

The solvent casting method was used to prepare nanocomposite films of PLA with Cloisite Na^+^, Cloisite 30B and Cloisite 20A nanoclays by Rhim et al. [73]. Among the clays used, Cloisite 20A showed the highest water vapor barrier performance with a BIF of 1.5 measured at 99% RH, but the tensile properties were sacrificed. The BIF values for the Cloisite Na^+^ and Cloisite 30B systems were 0.8 and 1.05, respectively. Zenkiewicz and Richert [74] used Cloisite 30B and Nanofil 2 nanoclays, poly(methyl methacrylate) (PMMA) and ethylene–vinyl alcohol copolymer as modifiers and polycaprolactone and poly(ethylene glycol) (PEG) as compatibilizers to fabricate a series of 27 PLA nanocomposite samples, and studied the effect on oxygen, water vapor and carbon dioxide permeability. Cloisite 30B samples improved the barrier properties much more than Nanofil clay—water vapor, oxygen and CO_2_ permeability decreased by 60%, 55% and 90%, respectively, at 5 wt% clay content. All the modifiers and compatibilizers decreased the CO_2_ transmission rate, whereas the oxygen and water vapor transmission rates were reduced only with the modifiers and not with the compatibilizers. The same group also studied the effect of blow molding ratio of PLA/MMT nanocomposite films containing (i) MMT, (ii) MMT with PMMA as modifier and iii) MMT and PEG as plasticizer on water vapor, oxygen and CO_2_ transmission. Among them, the MMT system showed the highest barrier performance with reduction in the transmission rates of water vapor, oxygen and CO_2_ by 40%, 40% and 80%, respectively. A further decrease by 10% to 27% was achieved by extrusion blow molding. The least permeable films were obtained at a blow molding ratio of 4 [75].

Koh et al. [76] prepared PLA nanocomposites with Cloisite 15A, Cloisite 20A and Cloisite 30B clays using the solution intercalation method. The Cloisite 30B system revealed exfoliated morphology in the TEM analysis and, consequently, outstanding gas (O_2_, CO_2_ and N_2_) barrier properties were obtained compared to the other clay systems and neat PLA. It was also found that the gas permeability decreased with decreasing kinetic diameter of the molecule: CO_2_ (3.3 Å) > O_2_ (3.46 Å) > N_2_ (3.64 Å), i.e., the films showed highest permeability for CO_2_ owing to its comparatively small size. The BIF values are shown in Table 1. Nanocomposites of aPLA and aPLA/polycaprolactone blends with organo-modified kaolinite (OM-kaolinite) were reported by Cabedo et al. [77]. Addition of OM-kaolinite drastically decreased the O_2_ permeability of aPLA (BIF 1.8, Table 1). The BIF of aPLA/PCL was 0.44 and the addition of OM-kaolinite was not as effective as in a neat aPLA matrix because of the effect of the interfaces in blend systems which provide a path for permeation. Trifol et al. [78] explored the synergistic effect of Cloisite 30B and cellulose nanofibers (CNF) on PLA barrier properties. A combination of 5 wt% Cloisite 30B and 5 wt% CNF showed a reduction of 90% in oxygen transmission rate and 76% in water vapor transmission rate compared to the neat PLA film. Even at a low filler content, 1 wt% of both materials, significant reductions in oxygen transmission rate, OTR (74%) and water vapor transmission rate, WVTR (57%), were achieved. However, in another study [79] 1 wt% Cloisite 30B in PLA showed only a 26% decrease in OTR and 43% in WVTR. Many groups have used Cloisite 30B as filler material in PLA matrix and demonstrated improved barrier properties, and the BIF values are shown in Table 1 [80,81,82,83,84]. Darie et al. [85] used nanoclays (Cloisite 93A modified with methyl dihydrogenated tallow quaternary ammonium and Dellite HPS, a hydrophilic smectite clay) varying in their degree of hydrophilicity to prepare PLA nanocomposites by melt processing. The O_2_ and CO_2_ transmission rates reduced by half in the Cloisite 93A matrix, and an even more drastic reduction occurred in the hydrophilic Dellite HPS matrix. Rhim [86] performed lamination of PLA films by using agar/κ-carrageenan modified Cloisite-Na^+^. The double layer and multilayer nanocomposite films showed a large decrease in OTR. Jalalvandi et al. [87] prepared nanocomposites of PLA/starch blend using unmodified MMT. The barrier properties of the nanocomposites were studied in terms of water uptake of the films. Neat starch films showed an uptake of 38%, whereas the nanocomposite with the highest clay loading of 7 wt% showed only 2% uptake. Othman et al. [88] varied the MMT clay content from 1 wt% to 9 wt% in the PLA matrix and obtained the best barrier performance at 3 wt% (BIF 1.5). In a similar study, Mohsen and Ali [89] varied the clay content and achieved best barrier properties at 4 wt%–6 wt% of nanoclay in a PLA matrix. A novel silver based organo-modified MMT (Bactiblock^®^ from Nanobiomatters, Spain) was used by Busolo et al. [90] to prepare antimicrobial PLA nanocomposite coating for food packaging. However, a reduction in the permeability of water vapor by only 20% was achieved at a clay loading as high as 10 wt%. Sengül et al. [91] used MMT clay modified with different organic modifiers (Table 1) and investigated the effect of clay modification and ratio on the barrier properties of PLA nanocomposites. The oxygen permeability decreased by 22% to 49% and the water vapor permeation by 46% to 80%. Chowdhury [92] investigated the effect of clay aspect ratio and the degree of dispersion on the barrier properties of PLA nanocomposites. Three different clays modified with the same organic modifier were used in the study. The trend in permeability depended on the aspect ratio, dispersion and the degree of disorder of the clays in the matrix. Jorda-Beneyto et al. [93] prepared PLA/MMT nanocomposite bottles by injection stretch blow molding and obtained decreased oxygen and water vapor permeability compared to the neat PLA bottle.

The best improvement in O_2_ permeability of a PLA/clay system to date was achieved with layer-by-layer (Lbl) technique. Svagan et al. [94] prepared transparent films of PLA/MMT nanocomposites using Lbl techniques that showed tunable O_2_ barrier properties (Figure 3). Very thin laminar multilayer structures of chitosan and MMT were constructed by an Lbl process (driven by electrostatic interactions) on extruded PLA films. Light transmittance analysis revealed high optical clarity for the coated PLA films, and TEM images showed well-ordered laminar structures of the bilayers. When 70 bilayers were used, the oxygen permeability coefficient of the coated PLA reduced by 99% and 96% at 20% and 50% RH, respectively. The data correspond to better oxygen barrier properties than PET at these humidity levels. Federico et al. [95] developed quadlayers (QL) and hexalayers (HL) of alternating branched poly(ethylene imine), Nafion and MMT on PLA thin film by Lbl technique. The oxygen permeability reduced by 98% and 97% in dry and humid conditions, respectively for 10 HL and QL layers, whereas the water vapor transmission reduced by 78%. HL films displayed efficient barrier properties than the QL films.

## 4. Mass Transfer in Polymers

Small molecules, such as O_2_, CO_2_, H_2_O, N_2_, permeate through a polymer membrane due to a gas chemical potential gradient through the membrane. The chemical potential difference acts as the driving force for the molecules to permeate from the high chemical potential side to the side of low chemical potential. The phenomenon of permeant transport in polymers is described using the solution–diffusion model. According to this model, the permeation in polymers consists of three steps, as depicted in Figure 4: (a) sorption of the permeant from the high concentration side onto the membrane/film surface, (b) diffusion of the permeant along the concentration gradient through the membrane and (c) desorption through evaporation from the low concentration surface of the membrane. Deviations from a gradient with a straight line can be observed when the permeating molecule interacts with the polymer and is categorized as non-Fickian diffusion, which is described by the diffusion–relaxation model [96,97].

Based on the assumption that the diffusion takes place in the x-direction of a flat membrane/film, the process is described by Fick’s first law of diffusion that gives the relationship between flux (*F*) and the concentration gradient (*dc*/*dx*) [96]:(1)$$F=-D\frac{dc}{dx}$$
where *D* is the diffusion coefficient, *c* is the concentration and *x* is the direction of the moving permeant. This equation is used in steady-state conditions, i.e., when the permeant concentration does not change with time. The flux at steady state is defined as the amount of the permeant (no. of moles or weight) passing through a surface of unit area (perpendicular to the flow direction) per unit time:(2)$$F=\frac{q}{At}$$
where *q* is the amount of the permeant, *A* is the membrane area and *t* is the time. At steady state, the permeant concentration is constant on both sides (just inside the material), *c*1 and *c*2 for high and low concentration, respectively, of the film. Therefore, Equation (1) can be integrated across the total thickness (*L*) of the membrane which gives:(3)$$F=D\frac{\left(c_{1}-c_{2}\right)}{L}$$

Equations (2) and (3) can be equated to get *q*:(4)$$q=D\frac{\left(c_{1}-c_{2}\right)At}{L}$$

For gases, it is convenient to measure the partial pressure (*p*) of the gas that is in equilibrium with the polymer rather than the concentration. Henry’s law is applied [96] at sufficiently low concentration and, when the interaction between the permeating molecule and the polymer is small:(5)c=Sp
where *S* is the solubility coefficient of the permeant in the polymer. Hence, assuming there is no interaction between the permeant and the polymer, Equation (4) can be expressed as:(6)$$q=DS\frac{\left(p_{1}-p_{2}\right)At}{L}$$
which, can be rearranged as:(7)$$DS=\frac{qL}{\left(p_{1}-p_{2}\right)At}$$

Equation (7) is nothing, but the permeability, *P*, of the permeant at steady state:(8)$$P=\frac{qL}{At\Delta p}$$

Therefore, from Equations (7) and (8) permeability can be expressed as the product of the diffusion coefficient *D* and the solubility coefficient *S*:(9)P=DS

*D* is the kinetic term (from Equation (1)) describing the mass flux of permeant through the film in response to a concentration gradient and *S* is the thermodynamic factor arising due to the interactions between the polymer and permeant molecules, which, is the ratio of equilibrium permeant concentration at the high concentration side of the film to the permeant partial pressure [98]:(10)$$S=\frac{c}{p}$$

Equations (7) and (8) are very simplistic and can be applied to penetrants in rubbery polymers which typically exhibit Fickian behavior at low concentrations. For glassy polymers, deviation from Fickian behavior can be observed due to the restricted chain mobility, leading to slow polymer chain reorganization in comparison to permeant-induced swelling [99]. As a consequence, a dual-mode sorption model is used to describe gas sorption in polymers at temperature below the glass transition temperature (*T_g_*) [100].

The unsteady state portion of the mass transfer permeation process is described by Fick’s second law, given by [101]:(11)$$\frac{dc}{dt}=\frac{d}{dx}\left(D\frac{dc}{dx}\right)$$

When *D* is position-, concentration- and time-independent, Equation (11) is expressed as:(12)$$\frac{dc}{dt}=D\frac{d^{2}c}{dx^{2}}$$

When there is a strong interaction between the polymer and the permeant, *D* becomes dependent on time, position and concentration and Equation (11) is solved using numerical methods [102].

### 4.1. Measurement of Mass Transport Properties

Two basic methods are used to determine the permeability of gases or vapors in polymer films: (a) isostatic and (b) quasi-isostatic methods [103]. In the isostatic method (continuous flow method), one side of the film is exposed to a constant concentration of the permeant and zero concentration is maintained on the other side. On the zero-concentration side, inert gas is purged to carry the permeant to the detector for quantification. In the quasi-isostatic method (lag-time method) constant permeant concentration is maintained on one side and the permeant is allowed to accumulate on the other side to a very low concentration of <5 wt% of the concentration on the feed side. The permeant from the accumulated side is removed at regular time intervals and quantified to generate a plot of permeant quantity vs time. By applying specified initial conditions (concentration throughout the film to be equal to zero) and specified boundary conditions with constant permeant concentration on the feed side and zero permeant concentration on the permeate side, a mathematical expression can be derived to describe the situation [104]:(13)$$q=\frac{Dc_{1}}{L}\left(t-\frac{L^{2}}{6D}\right)-\frac{2Lc_{1}}{\pi^{2}}\overset{\infty }{\underset{n=1}{\sum }}\frac{{\left(-1\right)}^{n}}{n^{2}}\exp\left(\frac{-Dn^{2}\pi^{2}t}{L^{2}}\right)$$

When steady state is reached, *t* becomes sufficiently large, i.e., *t→*ꝏ, then the exponential term in Equation (13) becomes negligibly small and hence the Equation reduces to:(14)$$q=\frac{Dc_{1}}{L}\left(t-\frac{L^{2}}{6D}\right)$$

A plot of *q* vs *t* gives a straight line with an intercept on the time-axis. The slope of the straight-line curve is the steady state flux (*F = DC/L*, from Equation (1)) and the intercept is the time-lag (*t_lag_*) (the intercept is an extrapolation from the straight-line curve to the time-axis, thus it is a shorter time to reach the steady state):(15)$$t_{lag}=\frac{L^{2}}{6D}$$

The diffusion coefficient *D* can then be calculated from the above Equation as:(16)$$D=\frac{L^{2}}{6t_{lag}}$$

This is a simple time-lag analysis and may result in errors when measuring diffusion coefficient in concentration-dependent cases where *t_lag_* may vary with pressure differences across the membrane. In this case, a concentration-averaged diffusion coefficient can be estimated from the plot of normalized permeant flux, i.e., ratio of the flux at time *t* to the flux at equilibrium (steady state) as a function of time. The diffusion coefficient can then be estimated using the relationship [105]:(17)$$D=\frac{L^{2}}{7.199t_{1/2}}$$
where *t*_1/2_ is the time required to reach half of the steady state value. The permeability coefficient can be calculated using Equations (7) and (8). There is also another method to determine *D*, whereby the equation:(18)$$\frac{Q}{Q_{\infty }}=\frac{4}{\sqrt{\pi }}\sqrt{\frac{l^{2}}{4Dt}}\exp\left(\frac{-l^{2}}{4Dt}\right)$$
is fitted to *Q/Q*_∞_ vs *t* curve using a simplex search algorithm [106]. *Q* is the flow rate at time *t* and *Q*_∞_ is the steady-state flow rate. Equation (18) can be obtained from dynamic flow rate permeation experiments [107,108]. Assuming Henry’s law is valid, the solubility, *S*, can be calculated using:(19)$$S=\frac{Q_{\infty }l}{Dp}$$

It is also possible to obtain *D* and *S* and then *P* from a gravimetric method, but it is not considered here [109].

### 4.2. Factors Affecting Mass Transport

One important factor affecting the mass transfer in polymers is the free volume of the polymer. Free volume holes are created due to Brownian motion and thermal perturbations of the polymer chains. During the sorption process, the permeant molecule occupies a free volume hole and then diffuses by short ‘‘jumps’’ into neighboring holes. It can also occur through gradual motion into a new hole that develops next to the first hole due to Brownian motion. The latter process is not really thermally activated since there is no barrier in energy to get across. Thus, the transport depends on the static free volume (number and size of the holes) and dynamic free volume (frequency of jumps). The static free volume is independent of the thermal motions of the polymer chains and is related to the permeant solubility, *S*, whereas the dynamic free volume is due to the segmental motions of the chains and is related to permeant diffusivity, *D*. The solubility coefficient *S* is related to specific free volume by [110]:(20)$$v_{sp}=v-v_{0}=\frac{S}{\rho_{gas}}$$
where *v_sp_* is the specific free volume, *v* is the specific volume, *v*_0_ is the occupied specific volume and *ρ_gas_* is the density of the gas. The fractional free volume, *v_f_*, is given by:(21)$$v_{f}=\frac{v_{sp}}{v}$$

Assuming that the holes are identical spheres arranged in a cubic lattice with lattice constant ‘*a*’, the average radius of the holes, *R* can be calculated by:(22)$$R=a\;\sqrt[{3}]{\frac{3v_{f}}{4\pi }}$$

The gas diffusivity depends on the dynamic free volume of the matrix, size of the gas molecules (molecular diameter, *dʹ*) and the velocity of the gas molecules (*u*) by [111]:(23)$$D=g\;d^{\prime }u\exp\left(-\frac{\gamma v_{0}}{v_{sp}}\right)$$
where, *g* is a geometric factor and *γ* is the overlap free volume factor, i.e., the degree to which more than one molecule can access the same free volume site. Therefore, after regrouping, the constants in Equation (23) become:(24)$$D=A\exp\left(-\frac{B}{v_{f}}\right)$$

The higher the fractional free volume, the larger will be the diffusivity. The dependence of solubility on *v_f_* is weaker than the diffusivity. Thus, permeability often follows a similar dependence on free volume as the gas diffusivity.

The effect of temperature on permeability, diffusivity and solubility is modeled using the Arrhenius equation [104]:(25)$$P=P_{0}\exp\left(\frac{-E_{p}}{RT}\right)$$
(26)$$D=D_{0}\exp\left(\frac{-E_{D}}{RT}\right)$$
(27)$$S=S_{0}\exp\left(\frac{-\Delta H_{s}}{RT}\right)$$
where *P*_0_, *D*_0_ and *S*_0_ are the pre-exponential factors, *E_P_* and *E_D_* are the activation energies for permeation and diffusion, respectively and Δ*H_S_* is the heat of dissolution of the permeant molecule in the polymer. Based on Equation (9), *E_P_* can be given as:(28)$$E_{p}=E_{D}+\Delta H_{s}$$

*E_D_* is always positive, Δ*H_S_* can be positive for light gases like H_2_, O_2_ and N_2_ and negative for condensable vapors like water, C_3_H_8_ and C_4_H_10_.

Other factors which affect the transport phenomenon include polymer chain structure (flexibility, polarity), crystallinity, chain orientation and packing, permeant solubility and humidity [28,104]

## 5. Modeling of Permeability of Polymer/Clay Nanocomposites

The mass transport mechanism in polymers containing platelet fillers (like nanoclays, graphene, etc.) is similar to that in semi-crystalline polymers. In semi-crystalline polymers, the content, shape and size of the crystals and the superstructure they form (spherulites, axialites) affect the transport properties. Thus, the crystals are considered as the gas-impermeable phase in an otherwise permeable amorphous matrix. There is, however, an important difference between the effects of crystals and impermeable platelets. It is only in special cases that the crystals are randomly dispersed in the amorphous matrix, e.g., in ultra-high molar mass polyethylene. Normally, the spherulitic structure gives rise to “dead-ends” at points where the crystals splay, and all amorphous parts are not necessarily reachable by the permeant [112,113]. The gas sorption in amorphous polymers at low to moderate uptake is given by Equation (10) (Henry’s law) and for semi-crystalline polymers it is given, assuming that all the amorphous parts are accessible by the permeant, by [114]:(29)$$S=S_{0}\left(1-\phi_{c}\right)$$
where *S*_0_ is the solubility coefficient of the amorphous phase and *ϕ_c_* is the volume fraction of the crystalline phase. For a “theoretically” 100% crystalline polymer, *S* = 0. In nanocomposites, the clay platelets are the non-permeable phase dispersed in the permeable polymer phase. The three main factors that influence the transport properties in clay/polymer nanocomposites are (a) the volume fraction of the nanoparticles (*ϕ*), (b) aspect ratio (*l/w*) of the platelets and (c) platelet orientation with respect to the direction of diffusion [45,51]. Incorporation of nano-platelets results in a decrease in the permeability of the polymer due to the permeant having to circumvent the platelets (leading to a tortuous diffusion path, or, in other words, a labyrinth effect) and this reduced permeability, represented as the ratio of composite permeability to the neat matrix permeability (*P/P*_0_) or the ‘relative permeability,’ is plotted as a function of the filler volume fraction (*ϕ*) to describe the transport properties in several models. A typical plot displays the nonlinear decay in (*P/P*_0_) with increasing filler volume fraction [33]. The volume fraction, which is the main input parameter in all mathematical models, can be calculated with [49]:(30)$$\phi =\frac{\frac{w_{np}}{\rho_{np}}}{\frac{w_{np}}{\rho_{np}}+\frac{1-w_{polymer}}{\rho_{polymer}}}$$
where *w_np_* and *ρ_np_* are, respectively, the weight fraction and density of the nanoparticles and *w_polymer_* and *ρ_polyme_*_r_ are the weight fraction and density of the polymer matrix. The main assumptions in most of the models are that the platelets have a regular geometry (thin rectangular or circular shaped platelets) and form an ordered array in space arranged either parallel to each other or display a distribution of orientation [45]. The average orientation is assumed to be at a particular angle to the direction of diffusion of the permeant molecules. Some of the important and common models can be grouped into three categories of spatial arrangement (i) parallel arrangement, (ii) random positioning and (iii) arrangements at an angle θ ≠ 90° and these are discussed below.

### 5.1. Periodic Arrangement of Parallel Nanoplatelets

A simple permeability model was proposed by Nielsen [115]. In this model the platelets are considered to have a rectangular shape with a finite length (*l*) and thickness (*w*) and are dispersed evenly in the polymer matrix with orientation perpendicular to the diffusion direction. The basic theory of the model is that the presence of impermeable platelets forces the permeant molecules to follow a longer diffusion path by traversing around the platelets. Therefore, this is also called the ‘tortuous path’ model, as shown in Figure 5.

The solubility coefficient, *S*, of this clay/polymer composite can be arrived at, from Equation (29) as:(31)$$S=S_{0}\left(1-\phi \right)$$
where *S*_0_ is the solubility coefficient of the neat polymer and *ϕ* is the volume fraction of the clay nano-filler. The diffusion coefficient, being influenced by the tortuous path, is given by:(32)$$D=\frac{D_{0}}{\tau }$$
where *D*_0_ is the diffusion coefficient of the neat polymer and *τ* is the tortuosity factor that depends on the platelet shape, aspect ratio and its orientation in the matrix. It is defined as:(33)$$\tau =\frac{d^{\prime }}{L}$$
where, *dʹ* is the distance that the permeant molecules must travel through the film in the presence of platelets and *L* is the actual distance the molecule would have traveled in the absence of platelets, i.e., thickness of the membrane. From Equations (9) and (31), we have;
(34)$$\frac{P}{P_{0}}=\frac{1-\phi }{\tau }$$

If ⟨N⟩ is the average number of platelets that the permeant molecule encounters during diffusion and if each platelet enhances the diffusion length by *l/2* on average, then the tortuous path length (prolonged diffusion length) is given by:(35)$$d^{\prime }=L+\langle N\rangle \frac{l}{2}$$

Since, $\langle N\rangle =\frac{L\phi }{w}$, the tortuosity factor, *τ* becomes:(36)$$\tau =1+\frac{l}{2w}\phi$$ Combining Equations (34) and (36) gives:(37)$$\frac{P}{P_{0}}=\frac{1-\phi }{1+\frac{\alpha }{2}\phi }$$
where *α* = *l/w* is the aspect ratio of the clay platelets. This is Nielsen’s equation which shows that the relative permeability decreases with increase in *α* and *ϕ* in the nanocomposite membrane [115]. However, it can be used as a rough estimate only up to a threshold limit in filler content, *ϕ* ≤ 10%, beyond which the particles may aggregate leading to increased permeation. The Nielsen equation was remarkably successful in validating the permeability reduction in many polymer systems. Figure 6 shows the predicted permeability decay curves for Nielsen’s model at different aspect ratios. However, it should be highlighted that incomplete exfoliation or orientation of the platelets and the occurrence of voids will result in systems deviating from the Nielsen model [115].

A second model where the resistance to diffusion arising from the tendency of the permeant molecule to get constricted in the slits (distance between two adjacent platelets) along with the contribution from the platelet length is given by Cussler et al. [116]. In this model, platelets are considered to be arranged parallel in multiple layers with a narrow-slit separation (*s*) between the platelets in each layer. In this case the following equation was derived:(38)$$\frac{P}{P_{0}}={\left(1+\frac{da}{s\left(a+b\right)}+\frac{d^{2}}{b\left(a+b\right)}+\frac{2b}{L}\ln\left(\frac{d}{2s}\right)\right)}^{-1}$$
where *L* is the film thickness and other parameters are as defined in Figure 7.

Here, the volume fraction and aspect ratio are given by:(39)$$\phi =\frac{da}{\left(d+s\right)\left(a+b\right)},\;\alpha =\frac{d}{a}$$

In this model, *d* is half the platelet length, and hence the aspect ratio is half that of the Nielsen model. Since the slit is considered to be very narrow, the second term was neglected, and the simplified expression of the relative permeability is given as:(40)$$\frac{P}{P_{0}}={\left(1+\frac{\alpha^{2}\phi^{2}}{1-\phi }\right)}^{-1}$$

This model predicts a rapid reduction in relative permeability at low volume fraction, as opposed to Nielsen’s model which requires high volume fraction or aspect ratio to achieve the same reduction in permeability.

### 5.2. Random Arrangement of Parallel Nanoplatelets

Brydges et al. [117] described the relative permeability considering random positioning of the parallel platelets in each layer and used a stacking parameter *γ′ = x/2d* to account for the deviation from periodicity, i.e., it defines the horizontal offset of each ribbon layer with respect to the platelet layer beneath it. A case of *γ′* = 1/2 is when the platelets in one ribbon layer are positioned at the center of the slit gaps of the layer underneath, and thus gives the lowest permeability. For very high aspect ratio, *α* > 100, this model gives:(41)$$\frac{P}{P_{0}}={\left(1+\frac{\alpha^{2}\phi^{2}}{1-\phi }\gamma^{\prime }\left(1-\gamma^{\prime }\right)\right)}^{-1}$$

In another case, Lape et al. [118] also considered platelets of the same aspect ratio arranged in a random fashion in the parallel ribbons. The reduced permeability is given by the product of reduced area and increased diffusion path length:(42)$$\frac{P}{P_{0}}=\left(\frac{A}{A_{0}}\right)\left(\frac{d^{\prime }}{L}\right)$$

The distance that the permeant has to diffuse through the nanocomposite films is given by:(43)$$d^{\prime }=L+\langle N\rangle \langle n\rangle$$

This equation is similar to Equation (35) except that *l/2* is replaced by ⟨n⟩, which is the average distance the permeant travels to reach the platelet edge. Using statistical considerations, *d′* is estimated to be:(44)$$d^{\prime }=\left(1+\frac{1}{3}\alpha \phi \right)L$$

The area available for diffusion is calculated by dividing the volume available for diffusion by the distance traversed to cross the membrane:(45)$$\frac{A}{A_{0}}=\frac{\left(V_{tot}-V_{np}\right)/d^{\prime }}{V_{tot}/L}$$
where *V_tot_* is the total volume of the membrane and *V_np_* is the volume of the nanoplatelets. Using Equation (44), the relative permeability is then given by:(46)$$\frac{P}{P_{0}}=\frac{1-\phi }{{\left(1+\frac{1}{3}\alpha \phi \right)}^{2}}$$

Fredrickson and Bicerano [119] modeled the case of circular shaped nanoplatelets with length 2*R* and thickness *w* having an aspect ratio *α = R/w*. Two situations were considered, as shown in Figure 8, (a) when the average distance between the platelets exceeds *R* due to low volume fraction and aspect ratio (*αϕ* << 1), i.e., in the dilute regime, the relative diffusivity is given by:(47)$$\frac{D}{D_{0}}=\frac{1}{1+\kappa \alpha \phi }$$
where *κ = π/ln α* and (b) in the semi-dilute regime when the circular disks overlap due to higher aspect ratios (*αϕ* >> 1), the relation is given by:(48)$$\frac{D}{D_{0}}=\frac{1}{1+\mu \alpha^{2}\phi^{2}}$$
where *µ* is a geometric factor.

Gusev and Lusti [120] developed periodic three-dimensional computer models containing a random dispersion of disk platelets in an isotropic matrix and solved the Laplace’s equation for the local chemical potential *µ*, (∇P(r).∇µ=0). The expression developed for the relative permeability is given as:(49)$$\frac{P}{P_{0}}=\exp\left[-{\left(\frac{\alpha \phi }{x_{0}}\right)}^{\beta }\right]$$

The values of *β* and *x*_0_ are 0.71 and 3.47, respectively [98].

In Figure 9, prediction curves for Nielsen, Cussler, Fredrickson and Bicerano and Gusev and Lusti models are compared for three different aspect ratios, *α* = 10, 100 and 1000.

It is observed that the predictions for the decrease in relative permeability is different for different models, particularly in the region of low *ϕ* values. To avoid this anomaly, *P/P*_0_ vs *αϕ* can be plotted [46]. Cussler and Gusev’s models predict dramatic decrease in *P/P*_0_ to almost zero permeability for *ϕ* ≥ 0.02 at very high aspect ratios. The plots also show that, at low aspect ratios, the models predict the need for a large volume fraction to achieve a significant decrease in permeability.

### 5.3. Platelet Arrangement at an Angle θ ≠ 90° to the Diffusion Direction

The main assumption in all the models discussed above is that the platelets are aligned perpendicular to the diffusion direction and hence the tortuosity is the highest. However, Bharadwaj [121] described the case where the platelets can be oriented at different angles (≠90°) with respect to the direction of diffusion. For describing this nonuniformity in alignment, Nielsen’s model was modified accordingly by introducing an order parameter which gives the degree of orientation of the platelets to the diffusion direction:(50)$$S^{\prime }=\frac{1}{2}\left(3\cos^{2}\theta -1\right)$$
where *θ* is the angle between direction of diffusion and the unit vector normal to the nanoplatelets’ large surface. When the platelets are oriented perpendicular to the direction of diffusion (i.e., *θ* = 0), then *Sʹ* = 1, whereas when platelets are oriented parallel to the diffusion direction (i.e., *θ* = *π*/2, then *Sʹ* = −1/2. For a random degree of orientation, *Sʹ* = 0. The modified Nielsen’s equation is then given by:(51)$$\frac{P}{P_{0}}=\frac{1-\phi }{1+\frac{\alpha \phi }{2}\frac{2}{3}\left(S^{\prime }+\frac{1}{2}\right)}$$

The case of *Sʹ* = 1 presents maximum tortuosity, and hence, the greatest reduction in relative permeability can be observed. The values of the order parameter for the different orientations are shown in Figure 10.

## 6. Model Validation for PLA/Clay Nanocomposites

Although a large body of literature is available describing the effects of two-dimensional clay sheets on reducing the water vapor permeability and gas permeability in PLA/clay nanocomposites, only a handful is available where the mathematical models have been successfully validated to account for the experimental results. Ray et al. [67] prepared PLA nanocomposites with organically modified (N-(coco alkyl)-N,N-[bis(2-hydroxyethyl)]-N-methyl ammonium cation) synthetic fluorine mica by melt extrusion using a twin-screw extruder. Films were prepared by compression molding at 190 °C. The WAXD and TEM analysis revealed intercalation of the clay platelets. However, the reduction in oxygen permeability with increasing clay concentration could not be explained by an intercalated nano-structure. Nevertheless, HRTEM revealed the co-existence of mixed intercalated and exfoliated structures that were found to be responsible for the improved oxygen barrier. Nielsen’s tortuosity model was found to match the experimental results well, which confirmed the presence of exfoliated mica sheets in large amounts in the matrix, with negligible role of the intercalated structures in the observed gas barrier properties.

Guo et al. [122] used two different modifications of Cloisite-Na^+^; (i) Cloisite 30B modified with bis(2-hydroxyethyl) methyl hydrogenated tallow quaternary ammonium cation and (ii) Cloisite- RDP modified with resorcinol di (phenyl phosphate) (RDP) and studied the oxygen barrier properties of the organically modified clay composites and compared them with those of the unmodified clay composite. The O_2_ barrier performance was explained using the work of adhesion (W_a_) parameter obtained from contact angle measurements. W_a_ essentially describes the strength of affinity between PLA and the clay sheets. Higher W_a_ values, indicating strong affinity, were obtained for PLA/Cloisite 30B, and the lowest value was observed for the PLA/Cloisite-Na^+^. The bulky tallow molecule in Cloisite 30B helped in forming exfoliated nanostructures in the PLA matrix that, in turn, demonstrated the best barrier performance of the three clays. On the other hand, the PLA/Cloisite-Na^+^, with less interfacial interactions, was shown to be a poor barrier film. The authors obtained best fit of the experimental permeability data with Nielsen’s model. The aspect ratios calculated from curve fitting of the Nielsen model were smaller than that observed by TEM, and the difference was attributed to the tilt angle (angle between platelet and the direction of diffusion).

Picard et al. [123] investigated the role of clay platelets (Nanofil 804; MMT modified with a dihydroxy methyl tallow quaternary ammonium cation) on the PLA crystallization, as well as the gas (O_2_ and He) permeability and established a crystallization-permeability relationship in PLA/clay nanocomposites for the first time. Melt compounding of PLA and OMMT was carried out using a mini-extruder to prepare the nanocomposites, followed by compression molding to make 100 µm thick films. The nanocomposites showed improved gas barrier properties at two different filler concentrations. A ca. 15% to 25% reduction in permeability/diffusivity was observed –a change that was higher than that reported by Ray et al. [66,67,68,69,70] for several PLA/OMMT systems. Nielsen’s tortuosity model was used to provide an accurate description of the experimental results for relative permeability and diffusivity of the nanocomposites. A mean clay aspect ratio of 24 calculated from the model curve was found to be in good agreement with that obtained from TEM micrographs. The presence of the OMMT platelets increased the crystallinity of the PLA by 46%, which decreased the O_2_ permeability in the annealed nanocomposite films. This permeability decrease induced by the increase in crystallinity was described well by the Maxwell equation [122].

Li et al. [124] prepared PLA/OMMT nanocomposites by solution intercalation and, later, coagulation in water. The coagulated solid was dried, and compression molded to form films. Experimental results of relative permeability of CO_2_ were found to follow the Nielsen model well at low clay loadings, but the model underestimated the permeability at higher clay content; the model predicted 60% reduction in the permeability of the clay-free material at a clay content of 3 wt%, whereas the corresponding experimental value was only 40%. This is because the model describes the system better in the dilute regime but is inaccurate in the semi-dilute regime. In their study, composites with >3 wt% OMMT loading belonged to the semi-dilute regime, and the theoretical permeability matched well with the measured permeability at 1 wt% and 3 wt% of clay loading. Nevertheless, the Cussler model still overestimated the permeability at higher clay loading of 7 wt% (theoretical permeability = 9% and measured permeability = 19%). The reason is the aggregation of silicate layers at higher clay concentrations leading to nonuniform dispersion of the layers/platelets and a decreased “effective” aspect ratio. According to Equation (40), a decrease in *α* will substantially increase the relative permeability. However, the Cussler model was found to give a better prediction of the system compared to other models. The Bharadwaj model fitting for *S′* = 0 was unsuitable at all clay loadings and was attributed to the uneven orientation of the silicate layers, corroborating with the observations from theTEM micrographs. The use of *S′* = 1 (platelets perpendicular to the diffusion direction) yielded the same description of the system as the Nielsen model.

PLA/poly(butylene succinate)/clay nanocomposites prepared by Bhatia et al. [125] by melt extrusion showed improved O_2_ barrier property with increasing clay content. However, the formation of clay stacks and nonuniform dispersion at high clay loading (>3 wt%) led to reduced tortuosity, and further improvement in barrier properties was negligible. Nanocomposite films prepared by compression molding could be described by the Bharadwaj model only up to 3 wt% clay, beyond which deviations from the model occurred because of the aforementioned clay agglomerates/nonuniform clay dispersion.

Tenn et al. [126] investigated the effect of the clay platelet hydration on the barrier properties of PLA/OMMT nanocomposites. The relative water and oxygen permeability results were fitted to the Bharadwaj model. However, the fitting for the water permeability was quite unsuitable. It was concluded that, apart from the aspect ratio and orientation of the clays, other parameters, such as interaction between silicate layers and water molecules, rigidity of the polymer chains in the vicinity of the clay layers, degree of crystallinity and percolation effects at the clay–polymer interface can interfere with the tortuosity concept, and thus make the tortuosity models less useable for prediction purposes. Nevertheless, the Nielsen–Bharadwaj model was found to be the best model to fit for the relative oxygen permeability experimental data of the PLA/OMMT nanocomposites. It was suggested that the tortuosity concept can be applied in a straightforward way to a gas-polymer system when there is no interaction between the diffusing gas molecules and the polymer matrix, whereas in the case of water or organic species, in addition to the tortuosity, different physical phenomena and chemical interactions can play a large role during the course of permeation, which can result in significant deviation from the expected tortuosity-based results.

## 7. Conclusions

Two-dimensional platelet/disk-shaped fillers (e.g., nanoclays) have been identified as the most effective nano-filler for increasing the gas barrier properties of polymers. These nanoparticles not only improve barrier properties of the polymer, but also improve mechanical properties and, often, the thermal stability owing to interfacial interactions with the polymer matrix. In this article, the commonly followed preparation methods for PLA/organoclay nanocomposites were elaborated which are solution intercalation, melt processing and in situ polymerization. The melt processing method is the most preferred route because of ease of implementation in industry. The barrier performance of PLA/clay nanocomposites with different kinds of nanoclay and with a vast variety of modifiers were reviewed to highlight the structure-property relationship, which varied from case to case. In general, the extent of exfoliation and stacking orientation of the nanoclays was found to be the most important factor affecting the barrier properties of PLA, where improvement by one or two orders of magnitude can be observed for fully exfoliated platelets. The individual clay platelets act as blockages and create tortuosity to the diffusing permeant molecules, and thus extend the diffusion path length and time. In many cases, they also reduce the solubility of the permeating gas molecules. Best barrier performance was found to be obtained through the Lbl technique. Although it is successful on the laboratory scale, the future success of this technique will depend on industrial implementation. The ability of the Lbl prepared clays to impart delayed diffusion is most useful in packaging and coating applications.

Some important mathematical models for estimating the relative permeability of polymer/organoclay nanocomposites have been described. The commonality among the models is the dependence of relative permeability on three factors: clay aspect ratio, volume fraction and the clay platelet orientation with respect to the direction of diffusion. Experimental validation of the models on PLA/clay systems has been studied only by a few groups and the results were reviewed in this article. Most of the models, Nielsen’s model in particular, were found to fit the data well at lower clay content. However, the models cannot be compared as the aspect ratios are different: some authors define aspect ratio as the width to thickness ratio while others define it as half width to thickness ratio. It can become more complicated because the degree of interaction between the polymer and the clay particles and the degree of delamination can be expected to vary. Nevertheless, with known aspect ratios of the clay, the simplest model proposed by Nielsen has proven to predict the relative permeability reasonably well. Another model which describes the tortuosity in polymer nanocomposites is the Fricke model that has been applied successfully in several composite systems, but has not so far been applied to PLA systems, although there is scope for in future studies [127].

## Figures and Tables

**Figure 1 polymers-12-01095-f001:**
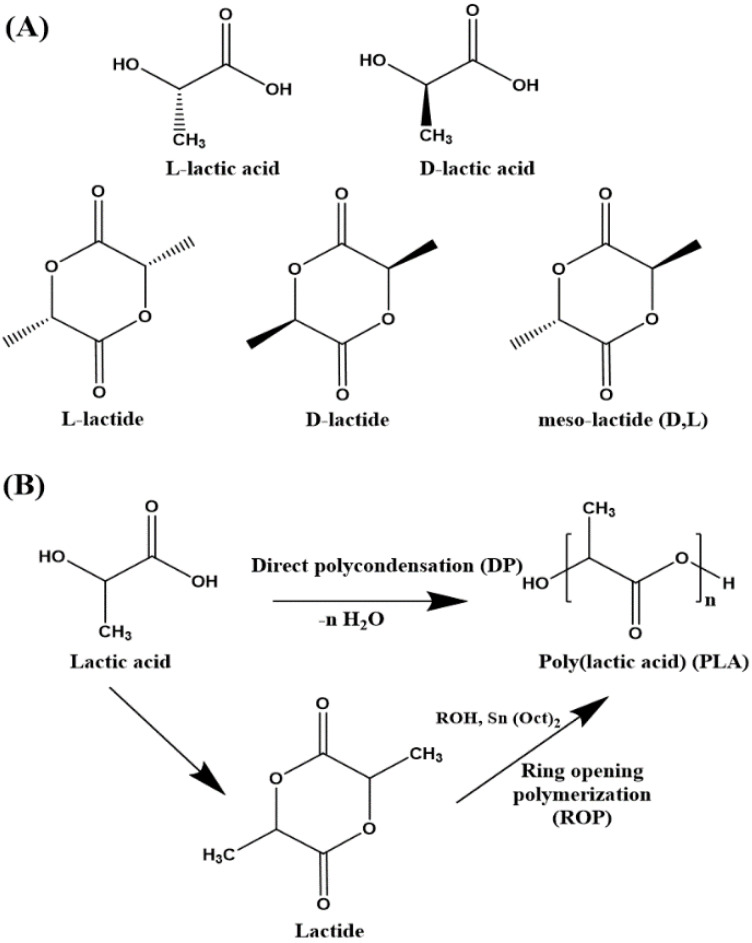
(**A**) Chemical structures of l-lactic acid, d-lactic acid, l-lactide, d-lactide and mesolactide, (**B**) Schematic representation of direct polycondensation (DP) of lactic acid and ring opening polymerization (ROP) of lactide.

**Figure 2 polymers-12-01095-f002:**
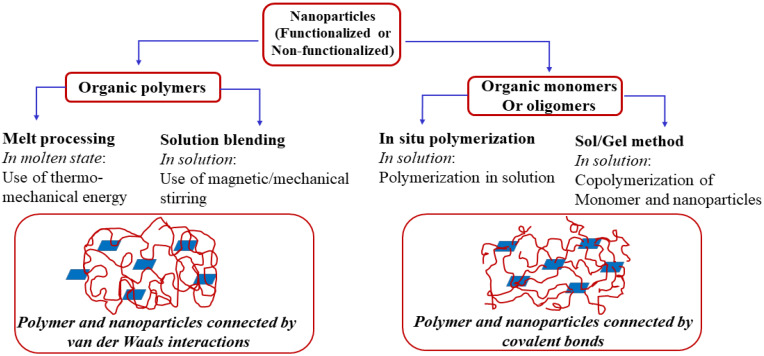
Outline of polymer nanocomposite preparation techniques.

**Figure 3 polymers-12-01095-f003:**
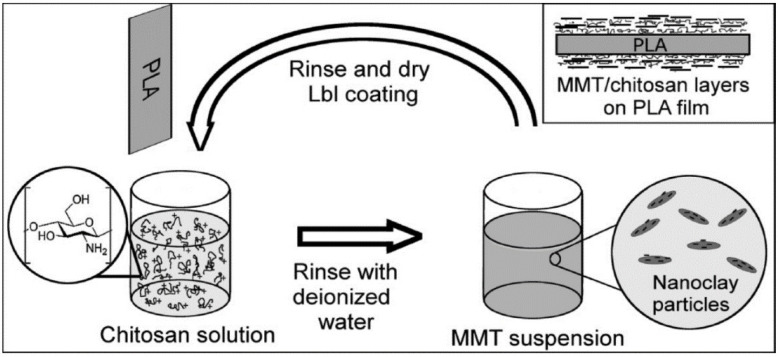
Schematic representation of layer-by-layer (Lbl) deposition of chitosan and montmorillonite (MMT) on extruded poly(lactic acid) (PLA) film. Reprinted with permission from Ref [94]. Copyright (2012) American Chemical Society.

**Figure 4 polymers-12-01095-f004:**
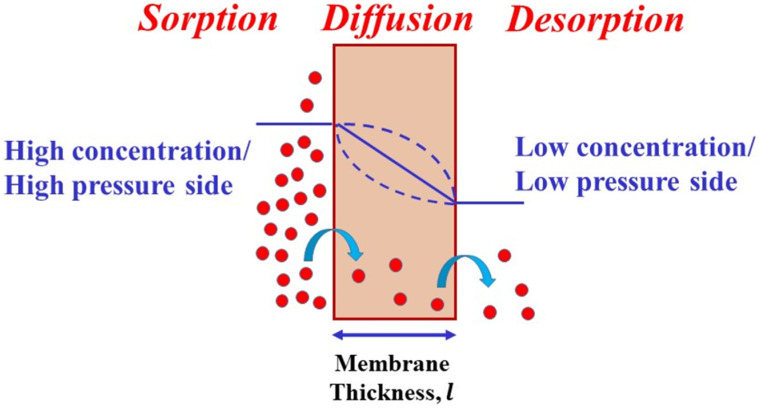
Schematic illustration of the solution diffusion model.

**Figure 5 polymers-12-01095-f005:**
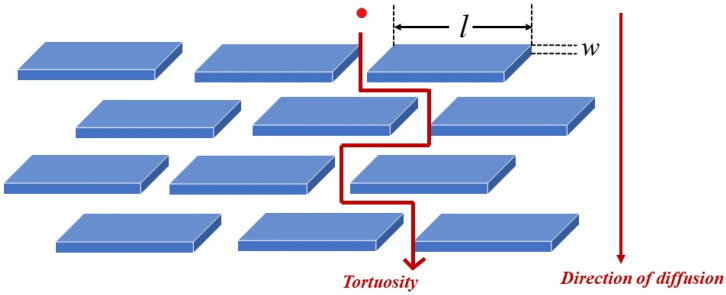
Schematic illustration of the tortuous path model.

**Figure 6 polymers-12-01095-f006:**
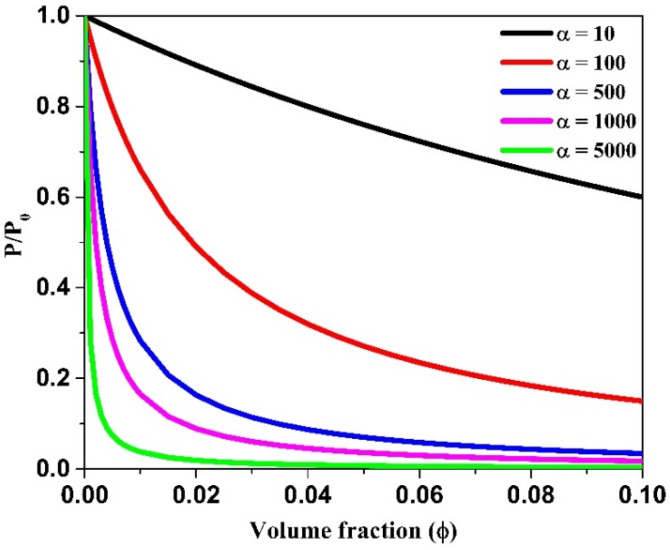
Prediction plot of Nielsen’s model at different aspect ratios.

**Figure 7 polymers-12-01095-f007:**
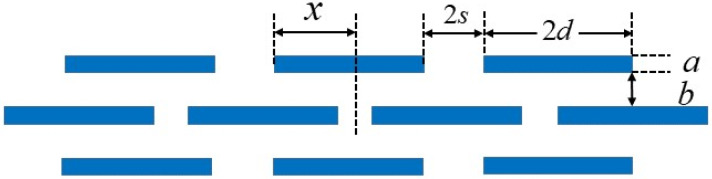
Ribbon arrangement of platelets.

**Figure 8 polymers-12-01095-f008:**
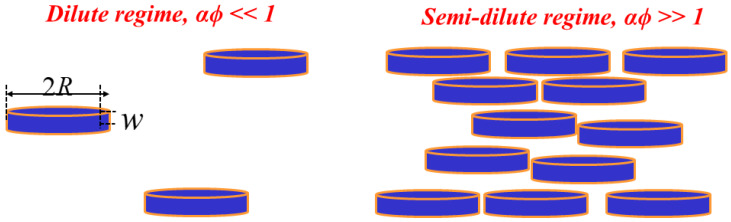
Schematic illustration of the dilute and the semi-dilute regimes of the oriented disk-shaped platelets.

**Figure 9 polymers-12-01095-f009:**
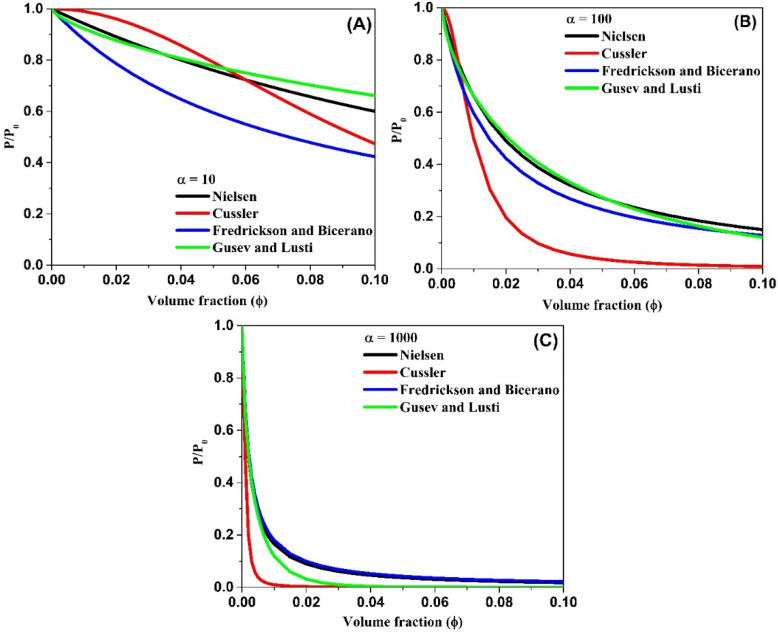
Prediction plots of relative permeability for different models at a fixed aspect ratio, (**A**) α = 10, (**B**) α = 100 and (**C**) α = 1000.

**Figure 10 polymers-12-01095-f010:**
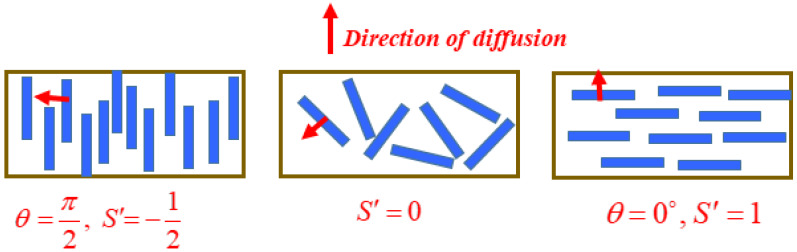
Different orientations of the platelets with the corresponding order parameter.

**Table 1 polymers-12-01095-t001:** Barrier improvement factors (BIFs) for PLA/clay nanocomposites.

Matrix	Nanoclay	Name and Formula of Organic Modifier	Penetrant	Clay Content	BIF	Ref
PLA	MMT	Dodecyltrimethyl ammonium,(Me)_3_(C_12_H_25_)N^+^	O_2_	10 wt%	2.3	[64]
	MMT	Hexadecyl ammonium,(C_16_H_33_)NH_3_^+^	O_2_	10 wt%	2.4	
	Cloisite 25A	Dimethyloctyl tallow amine(Me)_2_(C_8_H_17_)TN^+^	O_2_	10 wt%	2.3	
PLA	Smectite	Hexadecyltributyl phosphonium(C_4_H_9_)_3_(C_16_H_33_)P^+^	O_2_	4 wt%	1.7	[65]
PLA	MMT	Octadecyl ammoniumC_18_H_37_NH_3_^+^	O_2_	4 wt%	1.2	[66]
	MMT	Octadecyltrimethyl ammonium(Me)_3_(C_18_H_37_)N^+^	O_2_	4 wt%	1.1	
	Saponite	Hexadecyltributyl phosphonium(C_4_H_9_)_3_(C_16_H_33_)P^+^	O_2_	4 wt%	1.7	
	Synthetic fluorine mica (SFM)	Dipolyoxyethylene alkyl (coco) methyl ammonium(CH_2_CH_2_O)_x_H(CH_2_CH_2_O)_y_H(Me)R(coco)N^+^	O_2_	4 wt%	2.8	
PLA	SFM	N-(cocoalkyl)-N,N-[bis(2-hydroxyethyl)]-N-methyl ammonium(Me)(EtOH)_2_R(cocoalkyl)N^+^	O_2_	10 wt%	5.5	[67]
PLA	MMT	Octadecyltrimethyl ammonium(Me)_3_(C_18_H_37_)N^+^	O_2_	7 wt%	1.2	[68]
PLA	MMT	Octadecyl ammoniumC_18_H_37_NH_3_^+^	O_2_	7 wt%	1.5	[69]
PLA	SFM	N-(cocoalkyl)-N,N-[bis(2-hydroxyethyl)]-N-methyl ammonium(Me)(EtOH)_2_R(cocoalkyl)N^+^	O_2_	4 wt%	2.8	[70]
PLA	Cloisite 25A	Dimethyl hydrogenated tallow-2-ethylhexyl ammonium(Me)_2_(C_8_H_17_)(HT)N^+^	O_2_	5 wt%	1.7	[71]
			H_2_O	5 wt%	2.7	
aPLA	Kaolinite	Not disclosed	O_2_	4 wt%	1.8	[72]
	MMT	Not disclosed	O_2_	4 wt%	1.1	
PLA	Cloisite 20A	Dimethyl dihydrogenated tallow quaternary ammonium(Me)_2_(HT)_2_N^+^	H_2_O	5 pph	1.5	[73]
	Cloisite 30B	Methyltallow-bis-2-hydroxyethyl quaternary ammonium(Me)(CH_2_CH_2_OH)_2_(T)N^+^	H_2_O	5 pph	1.0	
	Cloisite Na^+^	Unmodified	H_2_O	5 pph	0.8	
PLA	Cloisite 15A	Dimethyl dihydrogenated tallow quaternary ammonium(Me)_2_(HT)_2_N^+^	CO_2_	0.8 wt%	2.0	[76]
			O_2_	0.8 wt%	1.4	
			N_2_	0.8 wt%	1.5	
	Cloisite 20A	Dimethyl dihydrogenated tallow quaternary ammonium(Me)_2_(HT)_2_N^+^	CO_2_	0.8 wt%	1.4	
			O_2_	0.8 wt%	1.1	
			N_2_	0.8 wt%	1.5	
	Cloisite 30B	Methyl tallow-bis-2-hydroxyethyl quaternary ammonium(Me)(CH_2_CH_2_OH)_2_(T)N^+^	CO_2_	0.8 wt%	2.0	
			O_2_	0.8 wt%	1.3	
			N_2_	0.8 wt%	2.0	
aPLA	Kaolinite	Not disclosed	O_2_	4 wt%	1.8	[77]
PLA	Cloisite 30B	Methyl tallow-bis-2-hydroxyethyl quaternary ammonium(Me)(CH_2_CH_2_OH)_2_(T)N^+^	O_2_	5 wt%	1.6	[78]
			H_2_O	5 wt%	2.1	
PLA	Cloisite 30B	Methyl tallow-bis-2-hydroxyethyl quaternary ammonium(Me)(CH_2_CH_2_OH)_2_(T)N^+^	O_2_	3 phr	1.5	[80]
PLA	Cloisite 30B	Methyl tallow-bis-2-hydroxyethyl quaternary ammonium(Me)(CH_2_CH_2_OH)_2_(T)N^+^	O_2_	1 wt%	187.0	[81]
			H_2_O	1 wt%	1.25	
PLA	Cloisite 30B	Methyl tallow-bis-2-hydroxyethyl quaternary ammonium(Me)(CH_2_CH_2_OH)_2_(T)N^+^	O_2_	2 wt%	1.6	[82]
			H_2_O	1 wt%	1.2	
PLA	Cloisite 30B	Methyl tallow-bis-2-hydroxyethyl quaternary ammonium(Me)(CH_2_CH_2_OH)_2_(T)N^+^	H_2_O	5 wt%	2.8	[83]
PLA	Cloisite 30B	Methyl tallow-bis-2-hydroxyethyl quaternary ammonium(Me)(CH_2_CH_2_OH)_2_(T)N^+^	O_2_	3 wt%	1.3	[84]
PLA	Cloisite 93A	Methyl dihydrogenated tallow quaternary ammonium(Me)(HT)_2_NH^+^	O_2_	3 wt%	2.0	[85]
			CO_2_	3 wt%	3.45	
	Dellite HPS	Not disclosed	O_2_	3 wt%	18.4	
			CO_2_	3 wt%	30.2	
PLA	Cloisite-Na^+^	Agar/κ-carrageenan	O_2_	5 wt%	516.0	[86]
PLA	MMT	unmodified	H_2_O	7 wt%	19.0	[87]
PLA	MMT	Not disclosed	O_2_	3 wt%	1.5	[88]
PLA	Clay name not mentioned	Not disclosed	O_2_	4 wt%	2.6	[89]
			H_2_O	6 wt%	3.1	
PLA	Ag-based MMT	Not disclosed	H_2_O	10 wt%	1.2	[90]
PLA	MMT	Dimethyldialkyl ammonium(Me)_2_(R)_2_N^+^	O_2_	10 wt%	2.0	[91]
			H_2_O	10 wt%	4.8	
		Aminopropyltriethoxysilane(CH_3_CH_2_O)_3_Si(C_3_H_6_)NH_2_	O_2_	10 wt%	1.5	
			H_2_O	10 wt%	2.7	
		Distearyldimethyl ammonium(C_18_H_37_)_2_(Me)_2_N^+^	O_2_	10 wt%	1.9	
			H_2_O	10 wt%	5.0	
		Hydrogenated tallow quaternary ammonium(HT)_4_N^+^	O_2_	10 wt%	1.7	
			H_2_O	10 wt%	2.3	
PLA	MMT(Southern clay)	Octadecyl ammoniumC_18_H_37_NH_3_^+^	O_2_	5 wt%	1.8	[92]
	MMT (Nanocor)	Octadecyl ammoniumC_18_H_37_NH_3_^+^	O_2_	5 wt%	1.3	
	SFM	Octadecyl ammoniumC_18_H_37_NH_3_^+^	O_2_	5 wt%	2.1	
PLA	MMT	Hexadecyltrimethyl ammonium(Me)_3_(C_16_H_33_)N^+^	H_2_O	4 wt%	1.6	[93]
			O_2_	4 wt%	1.7

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
