# Peer review of "A Review on Barrier Properties of Poly(Lactic Acid)/Clay Nanocomposites"

_polymers, 2020, doi:10.3390/polym12051095_

Round 1
Reviewer 1 Report
Dear Editor,
Dear Authors,
Very nice work.
The manuscript: “A review on barrier properties of poly(lactic acid)/clay nanocomposites” by Shuvra Singha and Mikael S. Hedenqvist reports very well on barrier properties of poly(lactic acid)/clay nanocomposites.
The MS is well written. The discussion is broad and comprises a wide range of conditions. The authors are able to introduce different technics and methods in a comprehensive way. The models are well explained. Everything is well cited and figures have the rights to be published.
Just one consideration:
In 120 references just 4 are from the last years. (2019-2020)
Would be possible to add newer references to have more studies that have been done in the last years?
The MS deserves to be published in Polymers after minor revision.
Author Response
Point 1: In 120 references, just 4 are from the last years (2019-2020). Would it be possible to add newer references to have more studies that have been done in the last years?
Response 1: We have gone through the literature thoroughly from scientific databases like Scopus and Scifinder. To the best of our knowledge we have included in the manuscript almost all the papers that were available till 2020 relating to barrier properties of PLA/clay nanocomposites. Only one new reference (which was earlier omitted) has been included in the revised manuscript.

Reviewer 2 Report
This paper reviews the application of PLA as a food packaging material. It highlights and briefly discusses the factors limiting food shelf-life and addresses the challenges. The authors talk about the existing preparation methods of PLA/clay nanocomposites and their effects on the final products. Barrier performance, mass transport in polymers and models for predicting permeability are further discussed. The article is well written and structured.
- I will suggest that if there are other applications of PLA aside food packaging, it should be clearly stated in the text and include why food packaging is of a major concern.
- What is the significance and effect of the arrangements and orientation of platelets on the permeability of PLA/clay nanocomposites?
- Can you elaborate on how nanocomposites improve the mechanical properties and thermal stability of polymers?
Author Response
Point 1: I will suggest that if there are other applications of PLA aside from food packaging, it should be clearly stated in the text and include why food packaging is of major concern.
Response 1: The major application of PLA (apart from food packaging) is in the biomedical and pharmaceutical industry where PLA is used for making biodegradable scaffolds for tissues, bioresorbable implants, surgical equipment, intravenal administration of antivirals, cardiovascular stents and controlled drug delivery. PLA is also used for making fibres in the textile industry and mulching materials for agriculture. We have included the applications in the manuscript.
The major source of plastic pollution that has afflicted humans, animals, land and water bodies has been identified to be the single-use plastics used in the food industry for storing and packaging solid and liquid food. These plastics (PE, PP, PET) are petroleum based and take years to degrade in the environment. Hence, research on biodegradable polymers such as PLA in the field of food packaging is of major concern.
Point 2: What is the significance and effect of arrangements and orientation of platelets on the permeability of PLA/clay nanocomposites?
Response 2: We have described in detail about the significance/effects of arrangements and orientation of the clay particles on the permeability in Sections 5.1 to 5.3 of the manuscript.
Point 3: Can you elaborate on how nanocomposites improve mechanical and thermal stability of polymers?
Response 3: Thermal and mechanical property enhancements of nanocomposites are obtained due to effective interfacial interactions between the polymer and the clay particles. Nanoparticles ought to be organically modified using an appropriate organic molecule that is compatible with the organic polymer. This ensures affinity of the polymer and the particles towards each other through ionic, hydrophobic or hydrogen bonding interactions. These interfacial interactions reduce the polymer chain mobility, thereby increasing the thermal and mechanical properties.
Reviewer 3 Report
The Review Paper entitled “A review on barrier properties of poly(lactic acid)/clay nanocomposites” prepared by Singha and Hedenqvist is an interesting investigation of the state of the art in the PLA nanocomposites. They reported some recent literature examples and some mathematical models. Moreover, Authors compared mathematical model and they discussed the results.
In my opinion the Paper deserves the publication. Anyway I suggest to improve the paper quality adding discussions on these two following points:
- Models reported take in account only the 2 dimensional aspects (length (l) and thickness (w)) of the platelets. I think that also the third dimension of the platelets could be important.
- Interaction with gas molecules was only discussed briefly in lines 737-740. I think that this argument could be discussed more in details.
Author Response
Point 1: Models reported take in account only the 2 dimensional aspects (length (l) and thickness (w)) of the platelets. I think that also the third direction of the platelets could be important.
Response 1: The primary assumption in all the models is that the clay platelets are arranged laterally (with the thickness (t) dimension normal to the direction of diffusion). This arrangement is assumed to increase the diffusion path length where the permeant has to circumvent the length (l) of the platelets. In such as arrangement, the third dimension does not have any role in obstructing the path. In fact the permeant glides through the width (w) i.e. the clay surface, during the course of permeation.
Point 2: Interaction with gas molecules was only discussed briefly in lines 737 – 740. I think this arrangement could be discussed more in detail.
Response 2: When there is little/no interaction between the permeating molecule and the polymer, the Nielsen’s tortuosity model can be applied directly and in a majority of cases the experimental data fit well with the model. In cases of strong interactions between the polymer and the permeant, the model may have to be modified to include parameters relating to polymer-permeant interaction forces.